# Minimizing Energy Usage and Makespan of Elevator Operation in Rush Hour Using Multi-Objective Variable Neighborhood Strategy Adaptive Search with a Mobile Application

Rojanee Homchalee [1] , Rapeepan Pitakaso [2] and Orawich Kumphon [1,*]

1 Applied Statistics Research Unit, Department of Mathematics, Mahasarakham University, Maha Sarakham 44150, Thailand; rojanee.h@msu.ac.th
2 Artificial Intelligence Optimization SMART Laboratory, Department of Industrial Engineering, Ubon Ratchathani University, Ubon Ratchathani 34190, Thailand; rapeepan.p@ubu.ac.th
* Correspondence: orawich.k@msu.ac.th; Tel.: +66-845112886

**Abstract:** The purpose of this study is to address two major issues: (1) the spread of epidemics such as COVID-19 due to long waiting times caused by a large number of waiting for customers, and (2) excessive energy consumption resulting from the elevator patterns used by various customers. The first issue is addressed through the development of a mobile application, while the second issue is tackled by implementing two strategies: (1) determining optimal stopping strategies for elevators based on registered passengers and (2) assigning passengers to elevators in a way that minimizes the number of floors the elevators need to stop at. The mobile application serves as an input parameter for the optimization toolbox, which employs the exact method and multi-objective variable neighborhood strategy adaptive search (M-VaNSAS) to find the optimal plan for passenger assignment and elevator scheduling. The proposed method, which adopts an even-odd floor strategy, outperforms the currently practiced procedure and leads to a 42.44% reduction in waiting time and a 29.61% reduction in energy consumption. Computational results confirmed the effectiveness of the proposed approach.

**Keywords:** vertical transportation system; optimization; multi-objectives variable neighborhood strategy adaptive search

**MSC:** 90-10; 90B06

## 1. Introduction

### 1.1. Background

The COVID-19 pandemic has underlined the need for maintaining a healthy environment in order to prevent the spread of infectious illnesses. One of the most critical issues in many public locations is managing the flow of people, especially in restricted spaces, such as elevators. Long lines and crowded elevators might promote the spread of diseases such as COVID-19, resulting in higher infection rates and potentially overwhelming healthcare systems [1]. The waiting areas for elevators have always been bottlenecks for crowd control, and with the pandemic, it has become an even more significant issue. Elevator waiting areas are often small and confined, and with the need for social distancing, the number of people allowed in these areas has decreased, leading to long waiting times. This increased waiting time can lead to crowds gathering in the waiting area, which could lead to the spread of the disease. Moreover, as people wait in these areas, they may touch the same surfaces, such as elevator buttons, which could be contaminated with the virus, and thus contribute to its spread, especially during rush hours [2–4]. Kwon and Kim [5] introduced a mobile application for managing lift passengers that can offer real-time information on lift occupancy, forecast the number of people waiting in the lift area using machine learning,

and guide passengers to their destination floors. It can also be used to limit the number of people allowed in the waiting area and provide touchless lift access via QR codes or other technologies. Several academic sources [6–9] have suggested using mobile applications to manage elevator passengers during non-rush hours. These apps can effectively reduce the number of people waiting in elevator areas, facilitating social distancing and preventing overcrowding in the elevators. Some of these apps employ algorithms to assign passengers to elevators based on their destination floors and can also monitor and control elevator capacity. Other features of these apps include touchless access, voice-activated controls, and real-time elevator status updates. The use of mobile apps in elevator management is seen as a promising approach for preventing the spread of COVID-19 while also enhancing elevator efficiency and safety [10]. These applications are mainly focused on reducing the chance of the disease spreading throughout working offices through elevator use, but none of these applications take into account energy consumption or lift load, such as the elevator's makespan. This research aims to develop an application to manage an elevator's passengers and the lift operation. The application will manage the passengers to reduce the chance of the disease spreading while keeping the lower operating cost of the lift.

The energy consumption of a lift is influenced by a number of factors, including elevator speed, weight, frequency of use, motor, and control system efficiency, building design, and maintenance and modernization [11–13]. In addition, the stopping pattern of the elevator can significantly impact its energy consumption. A lift that stops on every floor will consume more energy than one that only stops on every other floor [14]. Therefore, optimizing the stopping pattern of the elevator can lead to significant energy savings.

The optimization of elevator management has been explored in several studies using mathematical models and optimization techniques, such as integer programming, genetic algorithms, and multi-objective optimization [15–17]. While these studies have made significant progress in finding optimal elevator operation strategies, there are still some research gaps that need to be addressed. This paper aims to identify and address these gaps.

(1) While much research has been devoted to exploring the use of mobile applications and other technologies to manage elevator traffic and improve passenger safety, there remains a notable gap in the literature with regard to optimizing elevator energy consumption. This paper aims to address this gap by exploring optimal elevator operations to reduce energy usage.

(2) Despite a wealth of research on elevator systems, many studies have relied solely on simulations or theoretical models, neglecting the importance of real-world testing. To address this shortcoming, this paper presents a comprehensive analysis of elevator energy consumption, drawing on both theoretical models and empirical data from actual elevator systems.

(3) Despite the widespread use of elevators, there remains a lack of rigorous mathematical proof and empirical evidence regarding elevator-stopping patterns and their impacts on energy consumption. This paper seeks to fill this gap by presenting a detailed analysis of these factors, supported by both theoretical proofs and empirical data.

This research aims to develop an application for managing elevator passenger flow while minimizing energy consumption and makespan, the two main objectives of the study. To address these objectives, a multi-objective mathematical model will be presented, and effective heuristic approaches will be employed to identify optimal operating schedules for elevators, balancing the need to limit the spread of disease while maintaining efficient elevator operation. The resulting solutions offer important insights into the effective management of elevator traffic and may prove valuable in addressing the challenges posed by current and future public health crises.

The proposed mathematical model can be viewed as a particular instance of the combinatorial optimization problem, involving a multi-objective task capacity assignment problem (for assigning passengers to elevators) and a parallel machine scheduling problem. Both of these problems are recognized as NP-hard, rendering exact solutions infeasible for larger problem sizes [17]. In this study, the variable neighborhood strategy adaptive search

(VaNSAS) is utilized to tackle the problem and provide an effective solution approach. The proposed heuristic approach promises to offer valuable insights into the optimization of elevator traffic while minimizing energy consumption and makespan. VaNSAS was first published by Pitakaso et al. [18], and it has been successfully applied to solve many types of problems, such as the multi-echelon vehicle routing problem [19], the assembly line balancing problem [20], multiple parallel machine scheduling [21], and parameter optimization for friction stir welding [22]. In this study, we propose modifications to the variable neighborhood search adaptive search (VaNSAS) algorithm by introducing novel black box improvements that augment its solution-finding capability. Our modifications are based on the fundamental principles of local search methods, as presented in previous articles [23–25]. Additionally, we introduce new selection methods that are developed based on historical data obtained during the simulation of the algorithm. These improvements are expected to enhance the effectiveness of VaNSAS and contribute to the body of knowledge on optimization algorithms.

### 1.2. Related Literature

Elevators are a key factor for high-rise building success and popularity, with various factors to consider for an efficient vertical transportation strategy. Jetter and Gerstenmeyer [26] stated that super-high-speed elevators are not necessary for good travel times and handling capacity. The interrupted flow of elevator cabins enables short waiting times for continuous passenger flow. A vertical transportation system (VTS) is important for efficient building operation and occupant comfort, affecting project costs and transportation demands. Various techniques have been proposed to address VTS issues, such as optimization models developed by Koo et al. [27] and Markov chain Monte Carlo evaluations by Al-Sharif and Hammoudeh [28]. Kuusinen et al. [29] proposed linear programming to reduce costs during lift trips, while the vertical logistics planning system (VLPS) was developed to optimize the vertical logistics strategy at 22 Bishopsgate, London [30]. The efficient operation of elevators is necessary to minimize costs and waiting times. However, energy consumption and long queues may occur without proper problem-solving. To achieve overall efficiency, energy efficiency concepts and criteria should be implemented for architects, designers, planners, and customers when projecting the energy consumption of elevators from various manufacturers in Switzerland under real-world conditions [31].

No published research verifies that the stopping strategy affects the lowest consumption cost, making the optimal total cost under the energy consumption function with selective stopping strategies an interesting area of study. Other factors, such as worker assignment, capacity, and floor, are also crucial for minimizing the total cost of elevator movement. The integration of human management during rush hours during the COVID-19 pandemic era poses a new challenge for elevator scheduling. Due to the complexity of this problem, an exact method is not feasible, and a heuristic approach, such as VaNSAS, is necessary. VaNSAS is a flexible population-based heuristic that can solve a variety of problems, including this multi-objective decision problem. It is known for its fast computation, intensive search, and ability to escape local optima. VaNSAS is similar to other popular population-based heuristics, such as genetic algorithms [32], differential evolution algorithms (DE) [33], and particle swarm optimization [34].

VaNSAS can form different types of local search or improvement methods within the same algorithm. This allows for intensive or diverse searching as needed. Multi-objective VaNSAS (M-VaNSAS), a modified version of VaNSAS that can solve multi-objective decision problems, is used in this research. Khonjun et al. [35] implemented a mobile application and the Internet of Things to reduce the number of passengers waiting for an elevator and waiting time, aiming to reduce the spread of COVID-19, but the application requires passengers to download it, which is a weakness. Several articles have aimed to minimize elevator energy usage, including using different power distribution systems, implementing LQG regulators, and utilizing DC micro-grids and energy-efficiency devices [36–38]. This

study will focus on two strategies to reduce energy usage: optimizing elevator-stopping strategies and combining passenger assignment with scheduling.

The most recently published articles from the year 2023 presented five studies related to engineering and transportation [39–43]. Shi and Zhu [39] proposed a dual-phase control strategy for cargo transportation using a partial space elevator, which effectively balances multiple control objectives and achieves a stable equilibrium state. Chen et al. [41] investigated the association between the operation parameters and the finished product quality in large-scale vertical mill systems, proposing a multi-objective optimization framework that increases yield and specific surface area. Gu et al. [42] analyzed the seismic response of a 26-story-frame shear wall structure before and after the installation of elevators and optimized the design scheme. Maleki et al. [43] proposed a method to predict the risk of passengers being trapped in elevators during earthquakes, which could help with virtual drills and emergency plans. Finally, a further study proposed an efficient approach for elevator scheduling in smart buildings to reduce energy usage and enhance user experience, resulting in significant improvements in energy consumption and service time.

The rush hour elevator operating strategy has received limited attention in the literature. At this time, passengers tend to use the elevators simultaneously, leading to overcrowding in the elevator waiting for the area and multiple stops for the same elevator. These can result in increased waiting and energy consumption times. The study aims to create a mobile application for passengers to reserve elevators during rush hours. Passengers' input will be used to optimize energy usage and elevators' makespan. The multi-objective variable neighborhood strategy adaptive search (M-VaNSAS) will be used to optimize both objectives. The results will be sent to the elevator control system for automatic operation and to passengers for information on elevator arrival time.

This paper is organized as follows: Section 2 explains the mathematical formulations which illustrate the proposed problem. Sections 3 and 4 outline the designed methodology and its computational results, while Section 5 provides the conclusion and outlook of this paper.

## 2. The Modified Mathematical Model

The mixed-integer nonlinear program (MINLP) has been proposed for a variety of important scientific applications, encompassing diverse areas such as optimal management of electricity transmission [44], power system contingency planning and blackout mitigation [45], the design of water distribution networks [46], nuclear reactor recharging and servicing [47], and the reduction of environmental impact from utility plants [48].

The mathematical model was formulated by Pitakaso et al. [18] for the optimization of the total cost of vertical transportation in elevators, with a strategy that allows all elevators to stop on every floor that is requested by the user (passenger), was studied. In this article, two major factors, namely electric consumption ($\frac{f_1 f_2}{v} Mo_j M_{jk}$) and stopping ($P_{jl}$) parameters, such as odd/even and high/low floors, are added to the model as part of the elevator allocation strategy based on floor parity. This strategy leads to a reduction in energy consumption; however, in order to optimize the total cost, some of the workers have to use stairs for walking up or down. The modified mixed-integer nonlinear programming (MINLP) models can be expressed as

Strategy 1: To minimize the total cost of elevator movement with the energy consumption function and set the conditions for necessities of the odd/even floor stoppage policy for each half of the elevators.

Strategy 2: To minimize the total cost of elevator movement with electric consumption function and set conditions for the elevator to stop on the low/high floors of the building.

*Notation Used in MINLP for Assignment of the Elevator Problem*

Sets:
$I$    set of passengers/elevator users;
$J$    set of elevators;
$K$    set of rounds of passenger transportation in an elevator;
$L$    set of building floors.

Parameters:
$f_1$    average load factor (technology factor);
$f_2$    hoist height factor;
$e_1$    electric consumption cost when elevator moves up or down;
$e_2$    electric consumption cost when elevator stops (opening and closing the door);
$v$    speed in meters per second;
$L_i$    required floor of building as the stop for the $i$-th passenger;
$G_{il}$    $\begin{cases} 1 & ; \text{ the } i\text{-th passenger wants to stop on the } l\text{-th floor of the building} \\ 0 & ; \qquad\qquad\qquad \text{otherwise} \end{cases}$
$P_{jl}$    $\begin{cases} 1 & ; \text{ the } j\text{-th elevator stop on the } l\text{-th floor of the building} \\ 0 & ; \qquad\qquad\qquad \text{otherwise} \end{cases}$
$W_i$    weight of the $i$-th passenger;
$C_j$    capacity of the $j$-th elevator;
$Mo_j$    motor output of the $j$-th elevator (kilowatts);
$D$    time required for passenger boarding/alighting;
$R$    elevator travel speed (in minutes per floor);
$Q$    elevator usage duration limit (in minutes).

Decision variables:
$Y_{ijk}$    $\begin{cases} 1 & ; \text{ the } i\text{-th passenger is assigned to the } j\text{-th elevator in the } k\text{-th round} \\ 0 & ; \qquad\qquad\qquad \text{otherwise} \end{cases}$
$X_{jk}$    $\begin{cases} 1 & ; \text{ the } j\text{-th elevator in the } k\text{-th round is in use} \\ 0 & ; \qquad\qquad \text{otherwise} \end{cases}$
$T_{ijkl}$    $\begin{cases} 1 & ; \quad \text{the } i\text{-th passenger that is assigned to the } j\text{-th elevator} \\ 0 & ; \quad \text{in the } k\text{-th round stop in the } l\text{-th floor of the building} \\ & \qquad\qquad\qquad \text{otherwise} \end{cases}$
$B_{jkl}$    $\begin{cases} 1 & ; \text{ the } j\text{-th elevator in the } k\text{-th round stop in the } l\text{-th floor of the building} \\ 0 & ; \qquad\qquad\qquad \text{otherwise} \end{cases}$
$M_{jk}$    Maximum floors that the $j$-th elevator travels in the $k$-th round;
$N_{jk}$    Starting time of the $j$-th elevator in the $k$-th round;
$F_{jk}$    Finish time of the $j$-th elevator in the $k$-th round.

Objective functions:

$$Min\ Z^1 = \sum_{k=1}^{K}\sum_{j=1}^{J} e_1 \frac{f_1 f_2}{v} Mo_j M_{jk} + \sum_{l=1}^{L}\sum_{k=1}^{K}\sum_{j=1}^{J} e_2 B_{jkl} \tag{1a}$$

$$Min\ Z^2 = \max_{jk}\left(F_{jk}\right) \tag{1b}$$

Constraints:

$$\sum_{k=1}^{K}\sum_{j=1}^{J} Y_{ijk} = 1; \forall_i \tag{2}$$

$$X_{jk} \leq X_{j,k-1}; \forall_j, \forall_{k=2,\dots,K} \tag{3}$$

$$\sum_{j=1}^{J}\sum_{i=1}^{I} Y_{ijk} X_{jk} - \sum_{j=1}^{J}\sum_{i=1}^{I} Y_{ij,k-1} X_{j,k-1} \leq 1; \ \forall_{k=2,\dots,K} \tag{4}$$

$$Y_{ijk} \leq X_{jk}; \ \forall_i, \forall_j, \forall_k \tag{5}$$

$$M_{jk} \leq Max_i\left(Y_{ijk} L_i\right); \ \forall_j, \forall_k \tag{6}$$

$$T_{ijkl} \leq G_{il}Y_{ijk}; \ \forall_i, \forall_j, \forall_k, \forall_l \tag{7}$$

$$\sum_{l=1}^{L} \sum_{k=1}^{K} \sum_{j=1}^{J} T_{ijkl} = 1; \ \forall_i \tag{8}$$

$$B_{jkl} \leq P_{jl}X_{jk}; \ \forall_j, \forall_k, \forall_l \tag{9}$$

$$T_{ijkl} \leq B_{jkl}; \ \forall_i, \forall_j, \forall_k, \forall_l \tag{10}$$

$$\sum_{i=1}^{I} W_i Y_{ijk} \leq C_j X_{jk}; \ \forall_j, \forall_k \tag{11}$$

$$N_{j1} = 0; \ \forall_j \tag{12}$$

$$F_{j1} = N_{j1} + RM_{j1} + \sum_{l=1}^{L} DB_{j1l}; \ \forall_j \tag{13}$$

$$N_{jk} = \left(F_{j,k-1} + RM_{j,k-1} + D\right)X_{jk}; \ \forall_j, \forall_{k=2,...,K} \tag{14}$$

$$F_{jk} = N_{jk} + RM_{jk} + \sum_{l=1}^{L} DB_{jkl}; \ \forall_j, \forall_{k=2,...,K} \tag{15}$$

$$F_{jk} \leq Q; \ \forall_j, \forall_{k=2,...,K} \tag{16}$$

Equation (1a) is an objective function that comprises two energy consumption costs, namely the moving cost and the stopping cost, while the second objective (1b) is to minimize the makespan of the elevators. Equation (2) guarantees that each passenger is assigned to elevator $j$ in round $k$ no more than once. Equations (3) and (4) stipulate that elevator $j$ in round $k$ is only permitted to operate if it was operated in the previous round, $k-1$. Equation (5) illustrates that passenger $i$ cannot be assigned to an elevator that is not operated. As shown in Equation (6), elevator $j$ during round $k$ must ascend to the maximum floor assigned to any passenger it transports. Equation (7) is used to calculate the floor $l$ for elevator $j$ in round $k$, which has to stop as passenger $i$ is assigned to that elevator. Equation (8) stipulates that a customer can be assigned to exactly one $j$, $k$, and $l$. Equation (9) is modified to ensure that the elevator round will stop on the floor only when that elevator is in use, with conditions for the elevator to stop on even/odd floors or high/low floors of the building. Equation (10) enables passenger $i$ to get off the elevator on floor $l$ only when elevator $j$ in round $k$ has stopped on floor $l$. As per Equation (11), the maximum allowable weight for the left elevator cannot be exceeded by the total weight of passengers assigned to it. Equation (12) shows that the start time of elevator $j$ in round 1 is the loading in time of the passengers to all elevators, while Equation (13) shows that the finish time of elevator $j$ in round 1 is the start time of round 1 plus the total loading time of all floors that the elevator stops at, plus the transportation time from floor $l$ to floor $M_{jk}$. Equation (14) shows that the start time of elevator $j$ in round $k$ is the finish time of elevator $j$ in round $k-1$ plus transportation time from $M_{j,k-1}$ to floor $l$, plus the loading time of passengers in round $k$. Equation (15) shows that the finish time of elevator $j$ in round $k$ is the start time of round $k$ plus the total loading time of all floors that the elevator stops at, plus the transportation time from floor $l$ to floor $M_{jk}$. Equation (16) shows that all elevators must finish moving in the last round within the limited operating time. The parameters' details are displayed in Table 1.

**Table 1.** Parameters used in the MINLP model.

| Parameters | Value | Unit |
|---|---|---|
| Average load factor (technology factor) * | 0.35 | - |
| Hoist height factor * | 1 | - |
| Speed * | 2 | m/s |
| Electric consumption cost when the elevator moves up or down | 3.5088 | THB/kw |
| Electric consumption cost when the elevator stops (opening and closing the door) | 5 | THB/time |
| Moving speed of the elevator | 0.5 | min/floor |
| Time required for opening and closing the door | 0.2 | min/floor |
| Maximum working time of the elevator | 40 | min |
| Capacity of elevator | 900, 1000, 1100 | kg |
| Motor output of elevator * | 21, 25 | kw |
| Weight of passengers 1 to 20 | 45–91 | kg |

* The factors from Nipkow and Schalcher [31].

## 3. The Proposed Method

In this article, the multi-objective vertical transportation problem will be solved using the multi-objective variable neighborhood strategy adaptive search (M-VaNSAS), which was first proposed by Pitakaso et al. [19]. The four steps of the VaNSAS algorithm for single-objective optimization consist of: (1) generating an initial set of tracks, (2) executing the track touring process, (3) updating heuristic information, and (4) repeating steps (2)–(3) until the termination condition is reached. For the multi-objective, step 3 has been modified in order to select the correct track to be the parent track for the next iteration of step (2). The random weight sum method will be used as the criteria of the selection process, while the Pareto points are collected during the simulation execution. M-VaNSAS can be explained stepwise as follows:

### 3.1. Generate the Set of Initial Tracks

$NT$ number track will be uniformly and randomly generated using Equation (17). Each track has a dimension of $1 \times D$, where $D$ represents the number of passengers and is set to 10. The initial tracks used in this paper are the integer numbers, and they are randomly generated using Equation (17).

$$E_{ni1} = U(0, J) \tag{17}$$

$E_{nm1}$ refers to the value of track $n$ at position $m$ during the first iteration, with $n$ and $m$ representing the predetermined track and position indices, respectively. $I$ denotes the number of passengers, and $N$ represents the total number of tracks. $J$ is the number of available elevators. The other two sets of tracks are also randomly generated in the first iteration, named here as the set of best tracks (BT) and random tracks (RT).

$$B_{ni1} = U(0, J) \tag{18}$$

$$R_{ni1} = U(0, J) \tag{19}$$

$B_{nmt}$ is the collection of the $N$ highest-quality solutions obtained from iterations 1 to $t$, as per Equation (18), and $R_{nmt}$ is randomly selected through Equation (19). Equation (20) is used to update $E_{nmt}$, and Table 2 shows an example of four randomly generated tracks.

$$E_{ij,t+1} = E_{ijt} \implies \text{Track execution process using designed operators} \tag{20}$$

**Table 2.** Four initial tracks with 10 dimensions, and the building has five elevators.

| $n\backslash m$ | 1 | 2 | 3 | 4 | 5 | 6 | 7 | 8 | 9 | 10 |
|---|---|---|---|---|---|---|---|---|---|---|
| 1 | 5 | 2 | 3 | 3 | 2 | 1 | 4 | 3 | 1 | 5 |
| 2 | 1 | 4 | 2 | 1 | 4 | 4 | 5 | 4 | 3 | 1 |
| 3 | 3 | 1 | 2 | 5 | 4 | 2 | 2 | 2 | 5 | 3 |
| 4 | 3 | 4 | 3 | 5 | 4 | 4 | 2 | 3 | 1 | 3 |

The value in position $m$ of track $n$ represents the number of elevators that passenger $m$ will ride. The maximum number of values in position $m$ equals the maximum number of elevators that are available. For example, passengers 1, 2, 3, and 4 will take elevators 5, 2, 3, and 3, respectively. The passengers that are assigned to the same elevator will be assigned one by one until the capacity of the elevator in each round is full, then the next round of that elevator be separated by the traveling round of the elevator. The second objective will increase by 1 if a passenger takes the elevator that does not stop at their desired floor. Both objectives can be calculated using Equations (1a) and (1b).

*3.2. Track Touring Process*

Utilizing black boxes iteratively will enable the improvement of the quality of the initial solution and the continuous search for better solutions. A black box can be designed freely with no limit rule and their use also has no limit rule. In selecting the track, Equation (21) incorporates various components, including random-transit (RT), best-transit (BT), inter-transit (IT), scaling factor (SF), differential scaling factor (DSF), and restart (RS). A roulette wheel selection approach was employed to select the preferred black box for the track. The "Roulette Wheel Selection" technique is frequently employed in metaheuristic selection processes owing to its capacity to effectively identify individuals for subsequent generations while ensuring a higher likelihood of the selection of individuals that exhibit superior fitness values. The technique involves assigning a probability of selection to each member of the population based on their individual fitness value, followed by a stochastic selection process using a "roulette wheel" mechanism that assigns greater probabilities of selection to individuals with superior fitness. This approach is known for its intuitive nature, computational efficiency, and potential to expedite the convergence toward optimal solutions. The selection of the black box is controlled by Equation (21).

$$P_{bt} = \frac{FN_{b,t-1}+(1-F)A_{b,t-1}+KI_{b,t-1}+\rho\left|A_{b,t-1}-A_{t-1}^{best}\right|}{\sum_{b=1}^{B} FN_{b,t-1}+(1-F)A_{b,t-1}+KI_{b,t-1}+\rho\left|A_{b,t-1}-A_{t-1}^{best}\right|} \tag{21}$$

The probability of selecting the black box in iteration $t$ is represented $P_{bt}$, with $N_{b,t-1}$ indicating the total number of tracks that selected a black box in the previous iterations, and $A_{b,t-1}$ denoting the average objective value of all tracks that selected black box $b$ in all preceding iterations. In this context, $A_{t-1}^{best}$ represents the globally optimal solution discovered before iteration $t$, while $I_{b,t-1}$ denotes a reward value that increases by 1 if a black box identifies the optimal solution in the previous iteration but remains constant otherwise, and $B$ signifies the total number of black boxes. The scaling factor is denoted by $F$, where $F = 0.5$, and $K$ represents the parameter factor, where $K = 1$. Additionally, $\rho$ signifies a predetermined parameter set to 0.05, which has been determined through the preliminary test.

*3.3. The Black Box*

Let us define the set of tracks in a single iteration, where $B$ represents the number of tracks included in each iteration. Define set $A$ as the collection of tracks that selected black box $b$ and set $Z$ as the collection of tracks that did not, such that the sum of the number of

tracks in $A$ and $Z$ is equal to $B$. Let $m$ denote the track randomly chosen from the set of tracks $Z$, and let $n$ denote the track in the set of tracks $A$.

Random-Transit(RT)
$$E_{niq} = \begin{cases} E_{ni,q-1} & ; \quad R_{ij} \leq C \\ R_{niq} & ; \quad \text{otherwise} \end{cases} \tag{22}$$

Best-Transit(BT)
$$E_{ijq} = \begin{cases} E_{ni,q-1} & ; \quad R_{ij} \leq C \\ B_j^{gbest} & ; \quad \text{otherwise} \end{cases} \tag{23}$$

Inter-Transit(IT)
$$E_{niq} = \begin{cases} E_{ni,q-1} & ; \quad R_{ij} \leq C \\ E_{miq} & ; \quad \text{otherwise} \end{cases} \tag{24}$$

Scaling Factor(SF)
$$E_{niq} = \begin{cases} E_{ni,q-1} & ; \quad R_{ij} \leq C \\ R_{ni}E_{ni,q-1} & ; \quad \text{otherwise} \end{cases} \tag{25}$$

Differential Scaling Factor(DSF) $\quad E_{niq} = E_{riq} + F(E_{mit} - E_{nit}) \tag{26}$

Restart(RS) $\quad E_{niq} = R_{ni} \tag{27}$

$B_j^{gbest}$ is the set of tracks that gives the global best solution and the best solution obtained from black box $b$. $R_{ni}$ denotes a randomly generated number at position $i$ of track $n$. Equation (28) is utilized to execute the sub-iteration update of the position of $Y_{ni,q+1}$. The variable $q$ denotes the number of sub-iterations executed by black box $b$, a parameter set in advance. As stated in [10], the values of $C$ and $F$ are set to 0.5 and 0.7, respectively.

The evaluation of the solution that will iteratively search can be seen in Equation (28).

$$E_{ni,q+1} = \begin{cases} E_{niq} & ; f_{nt} \leq f_{nq} \text{ and update } f_{nt} = f_{nq} \text{ and } E_{nit} = E_{niq} \\ Y_{rjq} & ; \qquad\qquad\qquad \text{otherwise} \end{cases} \tag{28}$$

The objective function of track $n$ at iteration $t$ is denoted by $f_{nt}$, while $f_{nq}$ represents the objective function of track $n$ at sub-iteration $q$. The objective function used in Equation (28) is derived from Equation (29)

$$f_{iq} = w^1 f_{iq}^1 + w^2 f_{iq}^2 \tag{29}$$

when $w^1$ is the random weight for objective 1 and $w^2 = (1 - w^1)$ and $w^1 = U(0, 1)$. $f_{iq}^1$ and $f_{iq}^2$ is the objective function of objective $Z^1$ and $Z^2$, respectively. The Pareto front of the proposed objectives is collected separately from the objective function Equation (28) and the objectives shown in Section 2 are modified to Equation (30).

$$Max\ Z = -w^1 \sum_{t=1}^{T} \sum_{j=1}^{J} \sum_{i=1}^{I} BT_{ijt}X_{ijt} + \sum_{l=1}^{L} \sum_{i=1}^{I} C_l + w^2 \max_{jk} F_{jk} \tag{30}$$

The set of non-dominated solutions will be preserved using the Pareto front. Let $f^1(e_r)$ and $f^2(e_r)$ represent the objective function for objectives 1 and 2 of track $r$, respectively. Let $\mathcal{R}$ denote a set of feasible solutions. The decision vector set $e = (e_1, e_2, \ldots, e_i)$ represents the feasible solutions and $f^v(e) = (f^1(e), f^2(e), \ldots, f^V(e))$ represents the objective functions of the feasible solutions. An element $e$ dominates $e'$ if and only if $f^v(e) \leq f^v(e')$ for all $v \in \{1, 2, ..., V\}$.

Upon the completion of the M-VaNSAS algorithm and the acquisition of the Pareto front, the technique for order preference by similarity to the ideal solution (TOPSIS) can be employed for further analysis, as described in the works of Hwang and Yoon [49] or Tzeng and Huang [50]. Utilizing the technique for order preference by similarity to the ideal solution (TOPSIS), a promising set of parameters will be identified, followed by the construction of the normal decision matrix. This process involves transforming the various attributes' dimensions into non-dimensional attributes using Equations (31)–(37).

$$r_{lv} = \frac{x_{lv}}{\sqrt{\sum_{l=1}^{L} X_{lv}^2}} \tag{31}$$

$$U_{lv} = w_v r_{lv} \tag{32}$$

$$U_v^* = \{\max_L U_{lv} \text{ if } v \in V; \ \min_L U_{lv} \text{ if } v \in V^*\} \tag{33}$$

$$U_v' = \{\min_L U_{lv} \text{ if } v \in V; \ \max_L U_{lv} \text{ if } v \in V'\} \tag{34}$$

$$S_l^* = \sqrt{\sum_{v=1}^{V} (U_v^* - U_{lv})^2} \tag{35}$$

$$S_l' = \sqrt{\sum_{v=1}^{V} (U_v' - U_{lv})^2} \tag{36}$$

$$C_l^* = \frac{S_l'}{S_l^* + S_l'} \tag{37}$$

when $x_{lv}$ represents the objective function value of point $l$ for objective $v$, where $L$ is the number of points in the Pareto front. $V^*$ denotes the set of positive objective functions, while $V'$ represents the set of negative objective functions. The parameter $w_v$ is predefined and represents the weight associated with each objective function. $U^*$ ($U^* = \{U_1^*, U_2^*, \ldots, U_n^*\}$) and $U'$ ($U' = \{U_1', U_2', \ldots, U_n'\}$) denote the positive and negative ideal solutions, respectively. $S_l^*$ and $S_l'$ represent the separation measures for each alternative from both the positive and negative ideal solutions, which are utilized in computing the relative closeness to the ideal solution ($C_l^*$). The set of parameters that has a value of $C_l^*$ closest to 1 will be selected as the promising solution.

### 3.4. Revise the Probability of the Black Box

In this stage, certain heuristic information utilized in Equation (21) must be revised, and the variables requiring updates are presented in Table 3.

**Table 3.** Enumerated set of variables requiring iterative updates.

| Explanatory Variables | Adjustment Protocol |
| --- | --- |
| $N_{bt}$ | The cumulative count of tracks that selected black box $b$ from the first iteration up to iteration $t$. |
| $A_{bt}$ | $A_{bt} = \frac{N_{bt}}{T_{bt}}$ <br> when $T_{bt}$ denotes the aggregate cost incurred by all tracks that have selected the black box $b$, starting from iteration 1 up to iteration $t$. |
| $I_{bt}$ | $I_{bt} = I_{bt-1} + G$ <br> when $G = \begin{cases} 1 & ; \text{ if in iteration } t, \text{the global best solution is contained within black box } b \\ 0 & ; \qquad\qquad\qquad\qquad\qquad \text{otherwise} \end{cases}$ |
| $B_j^{gbest}$ | Update the track containing the globally optimal solution |
| $R_{ijq}$ | Select values at random for all positions across all tracks |

### 3.5. Pseudocode of M-VaNSAS

The previous steps are iteratively repeated until the termination condition is met. The M-VaNSAS's pseudocode is shown in Algorithm 1.

---

**Algorithm 1:** Multi-Objective Variable Neighborhood Strategy Adaptive Search (M-VaNSAS).

---

**Input:** Number of tracks (NT), Number of Parameters (D), Scaling factor (F),
Improvement factor (K), Value of C, Number of improvement box (IBPop)
**Output:** Best_Track_Solution
  **Begin**
    Population = Initialize Population (NT, D)
    IBPop = Initialize Informatiom IB (NIB)
    Encode Population to WP
    ***While** the stopping criterion is not met **do***
      ***For** i = 1: NT*
      *// selected improvement box by*
      *RouletteWheelSelection*
      *selected_IB = RouletteWheelSelection(IBPop)*
      *If (selected_IB = 1) Then*
        *new_u = RT (u)*
        *Perform RT*
      *Else if (selected_IB = 2)*
        *new_u = BT (u)*
        *Perform BT*
      *Else if (selected_IB = 3)*
        *new_u = IT (u)*
        *Perform IT*
      *Else if (selected_IB = 4)*
        *new_u = SF (u)*
        *Perform SF*
      *Else if (selected_IB = 5)*
        *new_u = RS (u)*
        *Perform RS*
      *Else if (selected_IB = 6)*
        *new_u = DSF (u)*
        *Perform DSF*
      *Perform Decoding Method, Weight Sum Method*
      *If (CostFuntion(new_u)) ≤ CostFuntion(Vi) Then*
        *Vi = new_u*
      *Update Pareto Front*
      ***End For Loop** // end update heuristics information*
    ***End While Loop***
  **End**
  Return Best_Vector_Solution

---

### 3.6. The Comparison Methods

In this research, a comparative analysis was conducted using the proposed techniques and a well-known metaheuristic algorithm called differential evolution (DE). The DE algorithm is inspired by nature and comprises four major steps: (1) generating an initial solution; (2) executing a mutation process; (3) performing a crossover process; (4) executing a selection process. For our problem, we modified the DE algorithm proposed by Srichok et al. [51]. The selection process in DE was revised using Equation (29).

### 3.7. Application and IoT for Elevator System Control Design

To evaluate the waiting lines, a HIKVISION DS-2CD1123G0E-I camera was installed, while IoT devices with infrared sensors were installed on each floor of the building to monitor the elevator's movement. The movement and operation of the elevators were controlled using a PLC system programmed with C language. Figure 1 illustrates the operational design framework of the proposed method.

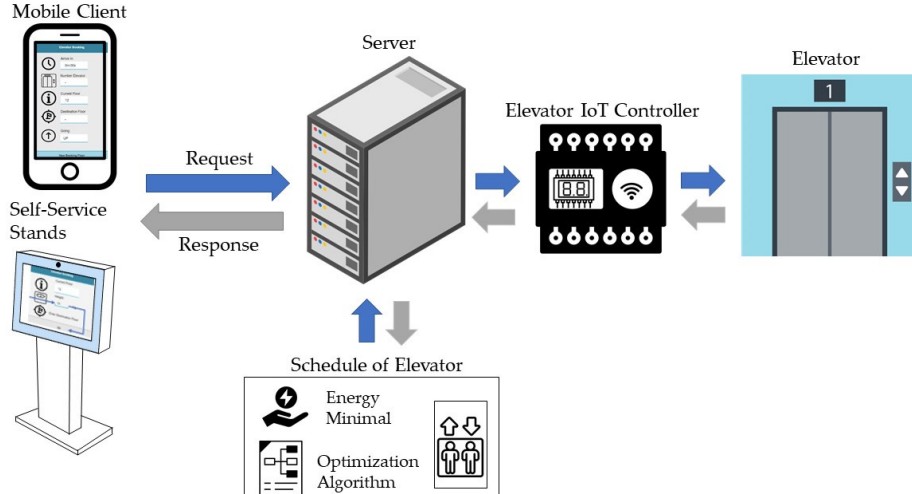

**Figure 1.** The proposed architecture designs.

The system architecture design when the mobile client requests the server, which records the daily elevator requests of users, is illustrated in Figure 1. Subsequently, the system queries all elevator bookings to schedule the use of elevators. To minimize energy consumption during the scheduling process, this system utilizes an optimization algorithm. The results of elevator booking are then relayed to each mobile client by the system. The proposed system includes an IoT controller for elevators, which transmits the real-time status information of elevators through the internet to the server for mobile client responses. The Android platform hosts the application, while PHP scripts are utilized for mobile client requests and responses on the server. The optimization algorithm, written in Python, is executed every time a mobile client makes an elevator booking. Moreover, the elevator booking system is installed on the touch screen panels of kiosks at the entrances of elevator zones to serve one-time visitors who do not have access to the booking system on their smartphones.

The user interface of the elevator booking system is depicted in Figure 2. To book an elevator, the user needs to click on "New Booking Floor", then provide their weight information and select the floor they want to go to. After confirmation by clicking on "Go", the system displays the resulting information, including the number of elevators assigned and the estimated usage time. Furthermore, real-time information on the current status of elevators on each floor is also displayed.

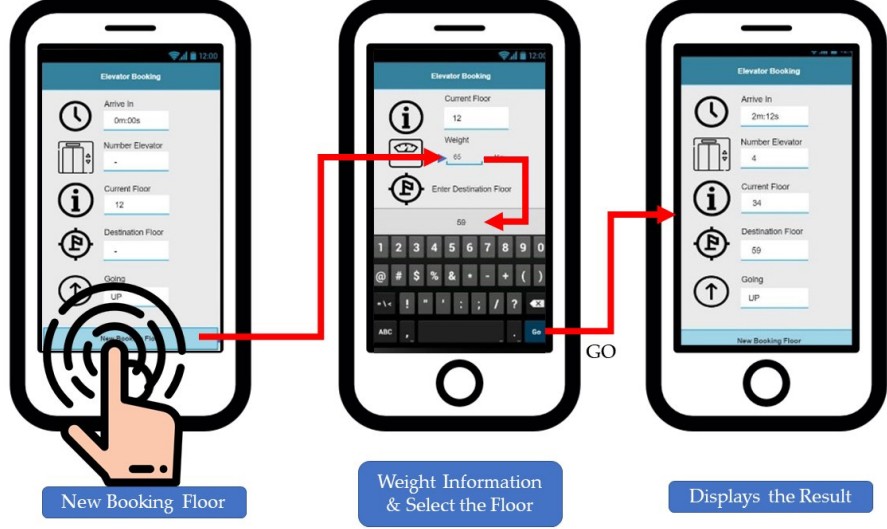

**Figure 2.** The proposed user interface.

## 4. The Computational Framework and Results

In this section, three groups are mentioned: (1) the computational results that reveal the effective stopping patterns of the elevator; (2) the computational results that reveal the effectiveness of M-VaNSAS compared with DE; and (3) the computational results of the designed system applied to the real case problem.

### *4.1. Best Elevator Stopping Strategy (Patterns)*

The threshold for tall buildings is 14 floors, or over 50 m in height, which is defined by the Council for Tall Buildings and Urban Habitats. In this section, the outcomes for the proposed problems, which are (1) the minimization of energy consumption and (2) the minimization of the makespan of the elevators, will be solved by LINGO; thus, the best strategy will be revealed subjected to different objectives. The sets, i.e., the number of passengers ($I$), the number of elevators ($J$), the number of rounds for moving up/moving down ($K$), and the number of floors ($L$), are varied for computation. $L$ is designed with two groups of buildings of normal and tall height. The varied sets are designed by $I = 20$, $J = 2, 4, 6, 8$, $K = 10$, and $L = 10, 15, 20, 25$. Two elevators are assigned, and each assigned elevator moves only one round. The workers are assigned to each elevator with a total weight of no more than its capacity. The number of stopping floors and the highest travel floor of the relevant problems are reported.

### 4.1.1. Minimize Energy Consumption

The optimal total costs for each strategy are shown as (285, 461, 603, 742) and (266, 352, 506, 602), respectively. Strategy 2 shows a better performance regarding the lower total cost than the results from strategy 1 (Figure 3), and the higher value of total cost is dependent on the number of floors. The extended computation experiments were also studied for the number of elevators ($J = 2, 4, 6, 8$) and are reported in Table 4. The total cost of energy consumption for each strategy is reported, from which can be seen that the majority of energy consumption costs from strategy 2 perform better than those of strategy 1. To execute the comparison of the two methods in this study a Wilcoxon signed-rank test was used. The Wilcoxon signed-rank test is often used in comparing the performance of two methods when the data are not normally distributed or when the sample size is small. The test is a non-parametric statistical test that does not make any assumptions about the underlying distribution of the data, making it a robust method for analyzing data that may not meet the assumptions of parametric tests, such as the *t*-test. Furthermore, the Wilcoxon signed-rank test is suitable for paired data, where each datapoint in a sample is directly related to a corresponding datapoint in another sample, such as when comparing the performance of two methods on the same dataset. This makes it a useful tool in various fields of research, including medical studies and engineering, where the performance of two or more methods needs to be compared to determine which method is more effective. The Wilcoxon signed-rank test indicated that energy consumption costs are better in strategy 2 ($Mdn2 = 413.687$) than in strategy 1 ($Mdn1 = 465.881$), $T = 18.00$, $Z = -2.585$, and $p < 0.009$. As a result, strategy 2 is suited to minimizing the objective function (1a) regarding energy consumption cost.

As can be seen from Table 4, the odd/even elevator system is more intricate and demands more resources to operate compared to the low/high system. There are specific reasons why the odd/even system is more costly. Firstly, in the odd/even system, each elevator can only stop on half of the floors in the building, requiring more elevators to cover all the floors than in the low/high system. This results in higher equipment, maintenance, and repair costs. Secondly, the odd/even system may require passengers to transfer elevators frequently to reach their desired floor, causing slower elevator operations, longer wait times, and frustrating experiences that can discourage building usage. Thirdly, to coordinate the odd/even elevator system, a more complex control system is required, with programmed elevators to alternate between odd and even floors and a control system to track the availability of elevators and the floors they stop at, driving up programming and

maintenance costs. Lastly, the odd/even elevator system has more elevators, resulting in more components requiring maintenance and repair. Malfunctions and breakdowns may also be more difficult to identify and fix, leading to longer elevator downtime, inconveniencing users, and reducing the building's efficiency. In conclusion, the odd/even elevator system, while seemingly a smart approach to distributing passenger traffic, is more expensive to operate due to several drawbacks. The low/high system is a more direct and efficient approach to elevator operation, which can result in cost savings in the long run.

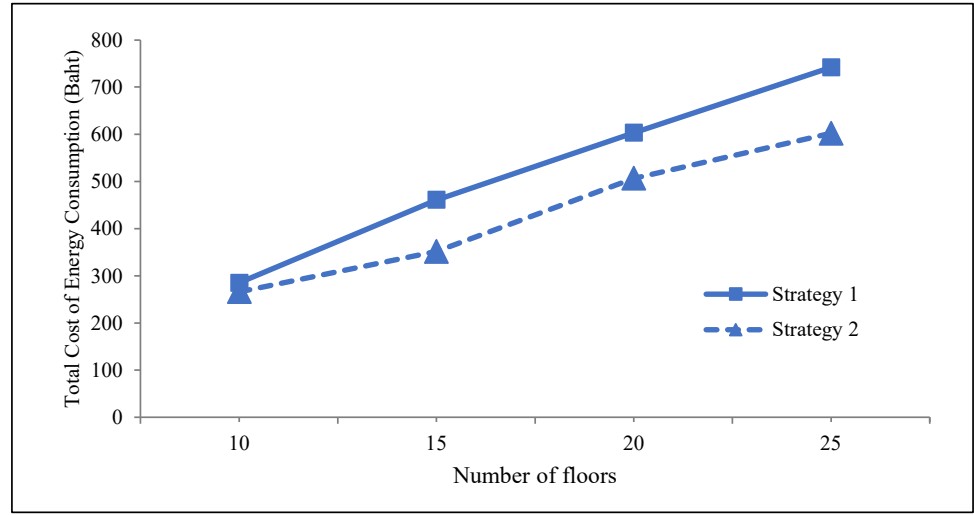

**Figure 3.** The minimization of the total cost of energy consumption against the number of floors ($I = 20$, $J = 4$, $K = 10$).

**Table 4.** Total cost of energy consumption for strategy 1 and 2 in different problem instances.

| Problem Instances | Number of Elevators (*J*) | Number of Floors (*L*) | Total Cost of Energy Consumption (THB) | |
|---|---|---|---|---|
| | | | Strategy 1 | Strategy 2 |
| 1 | 2 | 10 | 285.967 | 262.984 |
| 2 | 2 | 15 | 410.003 | 383.424 |
| 3 | 2 | 20 | 480.354 | 530.968 |
| 4 | 2 | 25 | 512.109 | 638.513 |
| 5 | 4 | 10 | 285.002 | 266.054 |
| 6 | 4 | 15 | 460.793 | 351.581 |
| 7 | 4 | 20 | 602.899 | 506.407 |
| 8 | 4 | 25 | 741.847 | 602.109 |
| 9 | 6 | 10 | 418.950 | 243.423 |
| 10 | 6 | 15 | 453.863 | 371.581 |
| 11 | 6 | 20 | 470.968 | 476.845 |
| 12 | 6 | 25 | 674.742 | 483.425 |
| 13 | 8 | 10 | 446.494 | 238.423 |
| 14 | 8 | 15 | 402.635 | 368.424 |
| 15 | 8 | 20 | 599.566 | 504.828 |
| 16 | 8 | 25 | 587.285 | 443.950 |
| | Median | | 465.881 | 413.687 |

### 4.1.2. Minimize Makespan

In this section, the majority of the results from strategy 1 reveal a lower makespan than those of strategy 2 (Table 5), but there is no significant difference between the two strategies ($Mdn1 = 5.0$, $Mdn2 = 5.4$, $T = 86.0$, $Z = 0.935$, and $p = 0.35$, respectively) according to the Wilcoxon signed-rank test. So, to find optimal values or minimize makespan could

be either strategy 1 or strategy 2. Figure 4 displays the graph depicting the correlation between the number of floors and makespan.

**Table 5.** Makespan for strategy 1 and 2 in different problem instances.

| Problem Instances | Number of Elevators ($J$) | Number of Floors ($L$) | Makespan (Mins) | |
|---|---|---|---|---|
| | | | Strategy 1 | Strategy 2 |
| 1 | 2 | 10 | 3.5 | 4.5 |
| 2 | 2 | 15 | 4.5 | 6.5 |
| 3 | 2 | 20 | 5.1 | 9.5 |
| 4 | 2 | 25 | 7.0 | 10.8 |
| 5 | 4 | 10 | 4.0 | 4.5 |
| 6 | 4 | 15 | 4.1 | 4.6 |
| 7 | 4 | 20 | 5.0 | 6.0 |
| 8 | 4 | 25 | 6.0 | 8.5 |
| 9 | 6 | 10 | 7.6 | 3.1 |
| 10 | 6 | 15 | 5.0 | 4.0 |
| 11 | 6 | 20 | 5.0 | 8.0 |
| 12 | 6 | 25 | 6.8 | 5.8 |
| 13 | 8 | 10 | 3.0 | 3.5 |
| 14 | 8 | 15 | 4.0 | 5.0 |
| 15 | 8 | 20 | 6.0 | 4.9 |
| 16 | 8 | 25 | 9.8 | 6.5 |
| Median | | | 5.0 | 5.4 |

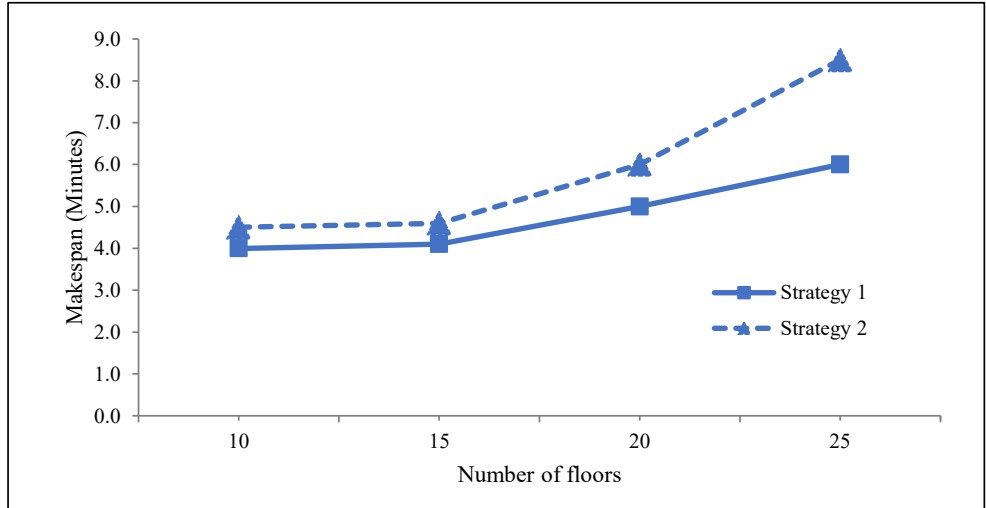

**Figure 4.** Makespan against number of floors ($I = 20$, $J = 4$, $K = 10$).

According to the computation results, the second strategy shows the best performance for the first objective when minimizing energy consumption; on the other hand, both strategies display the same ability for the second objective function, namely minimizing the makespan. The stopping pattern of an elevator can impact its makespan. A stopping pattern that divides the elevator's stops between odd and even numbered floors can result in longer travel times due to the elevator stopping on every floor. Alternatively, a pattern that stops at the floors on either the lower or higher half of the building and skips the floors in the other half can reduce the total operating time. Therefore, the low/high stopping pattern can be more efficient than the odd/even stopping pattern.

### 4.2. The Effectiveness of the M-VaNSAS and the Differential Evolution Algorithm (DE)

In this section, M-VaNSAS and DE were coded with Python and tested using an Intel®Core™ i5-2467M PC with a 1.6 GHz CPU. The testing was executed using the medium and large test instances, which LINGO v.11 cannot solve optimally. The number of elevators ranged from four to eight. The number of passengers ranged from 100 to 297. All cases were executed using M-VaNSAS and DE five times. The best solution among all runs was used as the representative of the methods. The stopping criterion for the simulation was the computational time, which was set to 45 min for all proposed methods. The first comparison was the number of Pareto points obtained from M-VaNSAS and DE when the number of iterations changed. The ratio of the number of points divided by the number of iterations is shown in Table 6.

**Table 6.** Comparing the Pareto ratio of DE and M-VaNSAS.

| Iteration | DE | | M-VaNSAS | |
| --- | --- | --- | --- | --- |
| | Number of Pareto Points | Ratio | Number of Pareto Points | Ratio |
| 200 | 280 | 1.40 | 340 | 1.70 |
| 500 | 601 | 1.20 | 891 | 1.78 |
| 800 | 933 | 1.17 | 1023 | 1.28 |
| 1000 | 1284 | 1.28 | 1506 | 1.51 |
| 1200 | 1490 | 1.24 | 1701 | 1.41 |
| 1500 | 1680 | 1.12 | 2014 | 1.34 |
| Average | 1044.67 | 1.24 | 1245.83 | 1.50 |

From the computational result shown in Table 6, it is revealed that M-VaNSAS found 78.18% more Pareto points than DE; thus, we can conclude that in finding more solutions, M-VaNSAS is better than DE. Figures 5 and 6 show the comparison of the Pareto front formed by VaNSAS and DE. Tables 7–9 show the computational results of DE and VaNSAS, in cases of $w^1 = 0.5, 0.7,$ and $0.3$, respectively. TOPSIS was used to reveal the solution. From Tables 7–9, we can see that in all cases, M-VaNSAS is able to generate a better solution than DE.

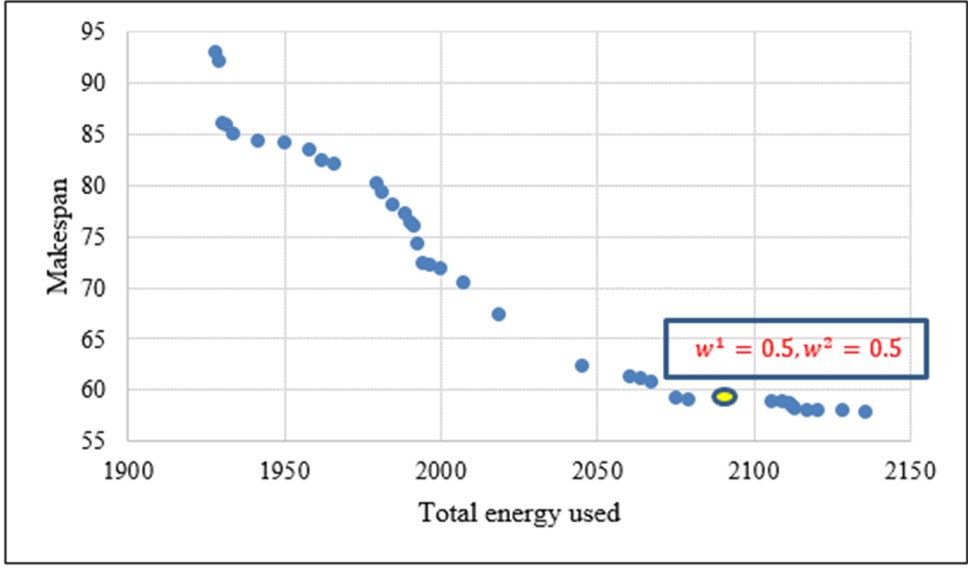

**Figure 5.** Pareto front of DE.

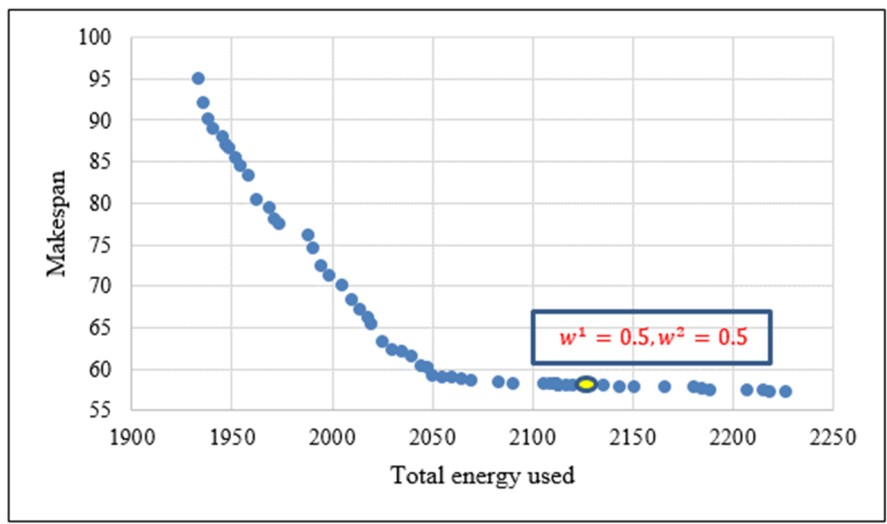

**Figure 6.** Pareto front of M-VaNSAS.

**Table 7.** Energy used and the makespan of the elevator using $w^1 = 0.5$ and $w^2 = 0.5$.

| Problem Instances | Number of Elevators (*J*) | Number of Floors (*L*) | Number of Passengers (*I*) | Total Energy Used | | Makespan | |
|---|---|---|---|---|---|---|---|
| | | | | DE | M-VaNSAS | DE | M-VaNSAS |
| 1 | 5 | 20 | 220 | 2331.36 | 2301.25 | 29.47 | 27.21 |
| 2 | 4 | 20 | 220 | 2398.60 | 2354.53 | 30.13 | 27.95 |
| 3 | 5 | 22 | 250 | 2460.08 | 2395.13 | 32.74 | 28.87 |
| 4 | 4 | 22 | 250 | 2538.98 | 2480.15 | 35.27 | 29.42 |
| 5 | 6 | 24 | 255 | 2687.09 | 2677.40 | 37.39 | 29.71 |
| 6 | 4 | 26 | 255 | 2852.98 | 2768.57 | 39.27 | 30.24 |
| 7 | 6 | 24 | 260 | 2952.72 | 2838.72 | 41.12 | 31.37 |
| 8 | 4 | 26 | 260 | 3218.64 | 3032.87 | 42.48 | 31.83 |
| 9 | 6 | 28 | 280 | 3350.38 | 3146.00 | 45.24 | 32.67 |
| 10 | 4 | 28 | 280 | 3406.57 | 3195.17 | 47.38 | 32.90 |
| 11 | 6 | 31 | 298 | 3467.74 | 3200.64 | 48.62 | 33.18 |
| 12 | 4 | 31 | 318 | 3525.17 | 3255.58 | 51.28 | 34.75 |
| | Median | | | 2932.53 | 2803.83 | 40.03 | 30.84 |

**Table 8.** Energy used and makespan of the elevator using $w^1 = 0.3$ and $w^2 = 0.7$.

| Problem Instances | Number of Elevators (*J*) | Number of Floors (*L*) | Number of Passengers (*I*) | Total Energy Used | | Makespan | |
|---|---|---|---|---|---|---|---|
| | | | | DE | M-VaNSAS | DE | M-VaNSAS |
| 1 | 5 | 20 | 220 | 2463.81 | 2395.89 | 28.48 | 27.18 |
| 2 | 4 | 20 | 220 | 2480.30 | 2446.30 | 29.18 | 27.45 |
| 3 | 5 | 22 | 250 | 2564.23 | 2487.85 | 31.23 | 28.01 |
| 4 | 4 | 22 | 250 | 2652.11 | 2577.13 | 34.12 | 28.91 |
| 5 | 6 | 24 | 255 | 2912.91 | 2823.66 | 36.45 | 29.30 |
| 6 | 4 | 26 | 255 | 2975.55 | 2878.45 | 38.58 | 30.12 |
| 7 | 6 | 24 | 260 | 3029.47 | 2975.40 | 39.13 | 30.58 |
| 8 | 4 | 26 | 260 | 3314.23 | 3108.94 | 40.48 | 30.97 |
| 9 | 6 | 28 | 280 | 3372.98 | 3191.74 | 42.33 | 31.23 |
| 10 | 4 | 28 | 280 | 3442.91 | 3233.47 | 43.81 | 31.28 |
| 11 | 6 | 31 | 298 | 3544.08 | 3256.00 | 45.74 | 33.31 |
| 12 | 4 | 31 | 318 | 3557.77 | 3327.62 | 49.76 | 32.91 |
| | Median | | | 3025.86 | 2891.87 | 38.27 | 30.10 |

**Table 9.** Energy used and makespan of the elevator using $w^1 = 0.7$ and $w^2 = 0.3$.

| Problem Instances | Number of Elevators (*J*) | Number of Floors (*L*) | Number of Passengers (*I*) | Total Energy Used | | Makespan | |
|---|---|---|---|---|---|---|---|
| | | | | DE | M-VaNSAS | DE | M-VaNSAS |
| 1 | 5 | 20 | 220 | 2282.68 | 2268.23 | 31.23 | 28.98 |
| 2 | 4 | 20 | 220 | 2370.72 | 2336.49 | 32.89 | 29.84 |
| 3 | 5 | 22 | 250 | 2415.40 | 2381.17 | 35.87 | 30.46 |
| 4 | 4 | 22 | 250 | 2496.11 | 2437.25 | 38.40 | 30.68 |
| 5 | 6 | 24 | 255 | 2670.72 | 2638.42 | 39.91 | 31.04 |
| 6 | 4 | 26 | 255 | 2824.04 | 2754.91 | 41.08 | 32.47 |
| 7 | 6 | 24 | 260 | 2940.08 | 2803.06 | 44.47 | 33.05 |
| 8 | 4 | 26 | 260 | 3174.30 | 3019.55 | 47.29 | 33.42 |
| 9 | 6 | 28 | 280 | 3301.25 | 3108.94 | 48.74 | 35.38 |
| 10 | 4 | 28 | 280 | 3376.26 | 3141.09 | 49.20 | 36.74 |
| 11 | 6 | 31 | 298 | 3397.62 | 3177.06 | 52.19 | 38.16 |
| 12 | 4 | 31 | 318 | 3508.72 | 3217.89 | 54.81 | 39.47 |
| Median | | | | 2896.49 | 2773.67 | 43.93 | 33.31 |

From the result obtained in Tables 7–9, we can see that M-VaNSAS has better total energy used (4.55%) than DE, as revealed from the average energy used for all the weights of $w^1$ and $w^2$. M-VaNSAS shows an average lower makespan than DE (29.61%). Thus, we can conclude that using M-VaNSAS provides a better solution than DE while using TOPSIS to evaluate the solution obtained from the Pareto analysis. M-VaNSAS provides better solutions in all differential weights of the objective functions. Figures 5 and 6 show the Pareto front of the DE and M-VaNSAS to uncover the real Pareto solutions obtained from DE and M-VaNSAS.

As can be seen from Figures 5 and 6, the Pareto front of M-VaNSAS can fill in the gap of the points better than DE. This result corresponds to that of the result shown in Table 6, where it is revealed that M-VaNSAS has a higher Pareto ratio than DE. This is the reason M-VaNSAS can fill the Pareto gap better than DE.

*4.3. Results of the Real-World Case Study*

We applied the designed application which used M-VaNSAS to decide the passenger assignment and the elevator's schedule for 30 days in an office building with 32 floors, six elevators, and 320 daily passengers. Rush hour operations using the designed application began at 7.30 and lasted until 8.15 am. The results collected for the 30 days compared the current situation and the results obtained using the application. The results are shown in Table 10.

**Table 10.** Compared results of the current situation and the proposed method.

| | Average Waiting Time (Minutes) | Average Number of Waiting Passenger (Person) | Energy Used (30 Days) (THB) | Average Makespan (Minutes) |
|---|---|---|---|---|
| Current situation | 22.45 | 41.63 | 149,871 | 45.98 |
| M-VaNSAS | 4.81 | 8.59 | 86,273 | 34.75 |

From Table 10, we can conclude that the M-VaNSAS uses less energy than the current situation (42.44%), and the makespan was reduced from 45.98 to 34.75 min. Moreover, the average waiting time of a passenger and the average number of passengers that are waiting in line was reduced by 78.57% and 79.36%, respectively. M-VaNSAS is the result of considerable study, experimentation, and refining targeted at establishing the most efficient and effective vertical transportation method to achieve a minimum waiting time, number

of waiting for passengers, energy utilized, and average makespan. It includes advanced technology, efficient processes, or specialized skills and knowledge, such as the designed lift booking application, IoT devices utilized as supporting tools to control the lift, and so on. The current practice, on the other hand, was developed or occurred over time without being subjected to a thorough review of its effectiveness. Such practices may be based on customs, habits, or a lack of resources and they may fail to take advantage of available technologies or information. Individuals can freely enter the waiting area; thus, the number of passengers waiting is undoubtedly far greater than the booking system of the produced software. When passengers utilize the program, their waiting time decreases dramatically because they are advised of their approximate arrival time. The number of people waiting has also fallen for the same reason. Since the effectiveness of M-VaNSAS to manage the elevator is considerably greater than that of the existing technique, the average makespan and energy required are reduced from those seen in current practice. Due to the uncontrollable arrival time of the passengers, the elevator will halt on every floor and must frequent the lowest floor. According to the newly produced system, the number of people waiting in the lift's waiting area is determined by the arrival time of people who are notified by the developed system. Although the newly designed system will coordinate the arrival time according to the lift's operation schedule, the number of waiting for people and the average waiting time of passengers in the new system will be significantly different from the current practice. This explanation has also resulted in the newly created system's lifespan and the energy consumption being significantly lowered in comparison to current practice.

## 5. Conclusions and Discussion

This study aims to address two major challenges that arise in the context of elevator operation: (1) the spread of epidemics, such as COVID-19, caused by long waiting times resulting from a large number of waiting for customers; and (2) excessive energy consumption due to the elevator usage patterns of various customers. The former is addressed through the development of a novel mobile application, while the latter is tackled by implementing two innovative strategies: (1) optimizing elevator-stopping strategies based on registered passenger information; (2) assigning passengers to elevators in a way that minimizes the number of floors that the elevators need to stop at. The optimization toolbox developed in this study integrates the mobile application with advanced search algorithms to solve the complex problem of elevator scheduling and passenger assignment. Specifically, the toolbox employs both exact methods and state-of-the-art metaheuristic algorithms, such as the multi-objective variable neighborhood strategy adaptive search (M-VaNSAS), to find the optimal method. The M-VaNSAS algorithm is a powerful tool that can handle multiple objectives simultaneously and can explore different search spaces efficiently. By integrating the mobile application with these advanced search algorithms, this study provides a novel and effective approach to optimize elevator operations and improve passenger experience, which is of great significance to the elevator industry and urban transportation management.

To ensure the effectiveness of the proposed approach in solving the complex multi-objective elevator scheduling problems, we designed and modified the multi-objective variable neighborhood strategy adaptive search (M-VaNSAS) algorithm. To enhance the search process, we incorporated various types of local search methods into the M-VaNSAS algorithm. These methods include random-transit (RT), best-transit (BT), inter-transit (IT), scaling factor (SF), differential scaling factor (DSF), and restart (RS), which allow the algorithm to explore different search spaces and converge on the optimal solution efficiently. The solutions obtained by the M-VaNSAS algorithm are then evaluated using the Pareto front and TOPSIS concepts, which enable the selection of the most optimal solutions based on their efficiency, accuracy, and robustness.

The results of the computational analysis demonstrate the superiority of the proposed approach, which incorporates an even/odd floor strategy, over the conventional approach

in terms of reducing both waiting time and energy consumption. Specifically, the proposed method achieved a reduction of 42.44% in waiting time and 29.61% in energy consumption. These results validate the effectiveness of the proposed approach and support its practical application in elevator systems. Furthermore, the computational analysis demonstrates the following findings:

(1) In terms of optimizing the elevator's stopping strategy to minimize energy consumption, strategy 2 outperforms strategy 1 by achieving an 11.2% improvement, while both strategies have the same makespan.

(2) The proposed M-VaNSAS algorithm yields a 4.55% reduction in energy consumption and a 29.61% reduction in makespans compared to the differential evolution (DE) algorithm, due to the more efficient local search procedure employed.

(3) The application and M-VaNSAS result in a significant reduction in energy usage by 42.44% compared to the current practice, along with a 78.57% and 79.36% reduction in the average passenger waiting time and waiting for line, respectively. Additionally, the proposed approach achieves a 24.42% reduction in makespan compared to the current situation in the building.

Numerous categories of elevators exist, including hydraulic elevators, traction elevators, machine-room-less (MRL) elevators, gearless elevators, double-deck elevators, and pneumatic elevators, which vary in their operating systems, energy usage rates, and suitability for different building types. For instance, hydraulic elevators are suitable for low-rise buildings, while traction elevators are appropriate for mid-rise and high-rise buildings. In this study, which focused on a building with 20–31 floors, traction elevators were exclusively employed. Traction elevators, designed for high-rise buildings, are capable of transporting passengers and goods swiftly and efficiently over long distances, thanks to the employment of hoists comprising steel ropes or belts to move the elevator car up and down the elevator shaft. This category of elevator typically boasts a high load capacity and can accommodate large numbers of passengers at once. It is worth noting that the mathematical model and M-VaNSAS suggested in this research can be utilized for other types of elevators. The parameter set required may include the operating cost, travel speed, and capacity of the elevators. Additionally, some constraints may be necessary, such as the operating regulations of specific lift categories. This represents an interesting research direction based on this study.

Many domains could be investigated in future studies to meet the goal of reducing energy consumption and increasing the efficacy of vertical transportation systems. One such path is to investigate the development and deployment of more capable and advanced control systems for elevators and escalators. This could include using machine learning algorithms to improve energy consumption and performance, as well as implementing smart sensors and monitoring systems to detect and adapt to changes in usage patterns. Another area of study might be the development of novel materials and technology for more energy-efficient and sustainable elevators and escalators. For example, new materials could be used to lower lift weight, leading to increased energy efficiency, or regenerative braking systems could be introduced to capture energy during operation. Finally, studies might focus on building design and construction to improve the performance of vertical transit networks. Using computer simulations to analyze the flow of people through a building and find areas where elevators and escalators could be placed most effectively to minimize energy consumption and increase efficiency is one example.

**Author Contributions:** Conceptualization, supervision, and methodology, R.P. software, validation, formal analysis, project administration, resources, and data curation, R.H. conceptualization, investigation, and writing—original draft preparation, review, and editing, O.K. All authors have read and agreed to the published version of the manuscript.

**Funding:** These research funds were supported by AIO laboratory, the FF65 research fund (Ubon Ratchathani University) and the Applied Statistical Research unit (Mahasarakham University). The facilities for this research were provided by Ubon Ratchathani University and Mahasarakham University.

**Data Availability Statement:** Applicable upon request.

**Conflicts of Interest:** The authors declare no conflict of interest.

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
