# Peer review of "Minimizing Energy Usage and Makespan of Elevator Operation in Rush Hour Using Multi-Objective Variable Neighborhood Strategy Adaptive Search with a Mobile Application"

_mathematics, doi:10.3390/math11081948_

Round 1

Reviewer 1 Report

This research aims to solve two main problems -viz. (1) when the number of waiting customers are

too high, it leads to longer waiting time which in turn lead to the spreading epidemics such as  COVID-19 and (2) energy consumptions are too high due to the pattern of the elevators from the point of view of various customers. The mobile application is designed to solve the first problem. The second problem be solved by two strategies which are (1) determine the optimal elevator’s stopping strategies subject to the registered passengers; (2) assign the passengers to the elevator in order to let the elevators stop in the lesser number of floors as possible. The mobile application platform is developed for the elevator’s passengers registered system and will be used as the input parameters to the optimization tool box. The exact method and the multi-objective variable neighborhood strategy adaptive search (M-VaNSAS) has been developed to find the optimal plan for the passenger assignment and elevator’s scheduling problem in order to optimize the energy usage and total operating time of all elevators. The computational results show that the best stopping strategies is even-odd floor strategy and the proposed method gives better solution than the currently practiced procedure—the reduction in the waiting time and the energy saved consequent to the implementation of the findings will be 42.44% and 29.61%, respectively.

The above is the edited abstract of the paper; it took me about 15 minutes to do it—first I should guess what the authors wanted to communicate through each sentence and then appropriate editing I had to make. Such preparation of manuscripts gives a bad impression in the readers.

The Even-Odd rule is already in vogue world over-- in road transport and other areas where it is meaningful. Therefore, this rule is not at all a novel introduction. The second objective of reducing energy consumption follows when the Even-Odd rule is implemented.

Therefore, I can not find anything thing new in the paper. While preparing manuscripts, especially scientific, the author(s) have to take at most care to see that it is easily readable (of course depending on the expertise of the reader). The present manuscript requires extensive editing as I demonstrated through the editing of its abstract.

Author Response

Dear Reviewer 1,

Thank you for your valueables comments and suggestions. Now, we replied and re-written as shown in the red colors via the attatchment file. We do hope this modification will meet your minimum requirement.

Regards,

Orawich Kumphon

Reviewer 2 Report

The paper has been written in an organized manner and supported with relevant and recent literature review. Congratulations!

A few comments to improve the paper:

·       Add reason for selection of roulette wheel for the track to select the preferred black box.

·       Add reason why the Wilcoxon signed-ranks test was used in comparing energy consumption costs between two strategies

·       Add more comment of Table 4 Total cost of energy consumption for strategy 1 and 2 in difference problem instances: why in certain instances Strategy 2 has higher total cost of energy.

·       Add more comment of Table 5 Makespan for strategy 1 and 2 in difference problem instances: why in certain instances Strategy 1 has higher value than Strategy 2.

·       Table 10. Compared result of the current situation and the proposed method shows a very significant improvement of the proposed method in terms of the average waiting time of the passenger (78.57%) and the average number of passengers that are waiting in line (79.36%). Please add more explanation how this could happen (how did the existing system work?)

·       A few grammatical errors need correction.

Author Response

Dear Reviewer 2,

Thank you for your valueable comments and suggestions. Now, we replied and re-written as shown in the red colors via the attatchment file. We do hope this modification will meet your minimum requirement.

Regards,

Orawich Kumphon

Reviewer 3 Report

Energy management for elevators in buildings and public places are essential in the sustainability of such systems. The authors have addressed an interesting issue on energy management. Although the work has potential for publication, addressing the following issues will help:

1. In practice, several categories of elevators are available but the authors should discuss the category used for analysis.

2. Authors needs to update references with 2003 journal articles.

3. Future research should be discussed in the conclusion.

4. Apart from makespan, compare results with other measures of efficiency.

Author Response

Dear Reviewer 3,

Thank you for your valueable comments and suggestions. Now, we replied and re-written as shown in the red colors via the attatchment file. We do hope this modification will meet your minimum requirement.

Regards,

Orawich Kumphon

Round 2

Reviewer 1 Report

The presentation of the manuscript has improved considerably in this revised version--it has become readable without much effort. However, further improvement is called for. As an example: line 11: The purpose of this study was to address two major issues: (1) the spread of epidemics --> The purpose of this study is to address two major issues: (1) the spread of epidemics. 
The authors must go through the manuscript thoroughly for corrections.

Author Response

Thank you for your suggestion, we revised our abstract to be a new version as shown in the blue color and underline.  

(The first revised manuscript was edited by MDPI with English editing ID: English-64458)

Abstract: The purpose of this study is to address two major issues: (1) the spread of epidemics such as COVID-19 due to long waiting times caused by a large number of waiting customers, and (2) excessive energy consumption resulting from the elevator patterns used by various customers. The first issue is addressed through the development of a mobile application, while the second issue is tackled by implementing two strategies: (1) determining optimal stopping strategies for elevators based on registered passengers and (2) assigning passengers to elevators in a way that minimizes the number of floors the elevators need to stop at. The mobile application serves as an input parameter for the optimization toolbox, which employs the exact method and multi-objective variable neighborhood strategy adaptive search (M-VaNSAS) to find the optimal plan for passenger assignment and elevator scheduling. The proposed method, which adopts an even-odd floor strategy, outperforms the currently practiced procedure and leads to a 42.44% reduction in waiting time and a 29.61% reduction in energy consumption. Computational results confirmed the effectiveness of the proposed approach.
